# SYMPLECTIC ODE-NET: LEARNING HAMILTONIAN DYNAMICS WITH CONTROL

**Yaofeng Desmond Zhong**[*]
Princeton University
y.zhong@princeton.edu

**Biswadip Dey**
Siemens Corporate Technology
biswadip.dey@siemens.com

**Amit Chakraborty**
Siemens Corporate Technology
amit.chakraborty@siemens.com

## ABSTRACT

In this paper, we introduce Symplectic[1] ODE-Net (SymODEN), a deep learning framework which can infer the dynamics of a physical system, given by an ordinary differential equation (ODE), from observed state trajectories. To achieve better generalization with fewer training samples, SymODEN incorporates appropriate inductive bias by designing the associated computation graph in a physics-informed manner. In particular, we enforce Hamiltonian dynamics with control to learn the underlying dynamics in a transparent way, which can then be leveraged to draw insight about relevant physical aspects of the system, such as mass and potential energy. In addition, we propose a parametrization which can enforce this Hamiltonian formalism even when the generalized coordinate data is embedded in a high-dimensional space or we can only access velocity data instead of generalized momentum. This framework, by offering interpretable, physically-consistent models for physical systems, opens up new possibilities for synthesizing model-based control strategies.

## 1 INTRODUCTION

In recent years, deep neural networks (Goodfellow et al., 2016) have become very accurate and widely used in many application domains, such as image recognition (He et al., 2016), language comprehension (Devlin et al., 2019), and sequential decision making (Silver et al., 2017). To learn underlying patterns from data and enable generalization beyond the training set, the learning approach incorporates appropriate inductive bias (Haussler, 1988; Baxter, 2000) by promoting representations which are *simple* in some sense. It typically manifests itself via a set of assumptions, which in turn can guide a learning algorithm to pick one hypothesis over another. The success in predicting an outcome for previously unseen data then depends on how well the inductive bias captures the ground reality. Inductive bias can be introduced as the prior in a Bayesian model, or via the choice of computation graphs in a neural network.

In a variety of settings, especially in physical systems, wherein laws of physics are primarily responsible for shaping the outcome, generalization in neural networks can be improved by leveraging underlying physics for designing the computation graphs. Here, by leveraging a generalization of the Hamiltonian dynamics, we develop a learning framework which exploits the underlying physics in the associated computation graph. Our results show that incorporation of such physics-based inductive bias offers insight about relevant physical properties of the system, such as inertia, potential energy, total conserved energy. These insights, in turn, enable a more accurate prediction of future behavior and improvement in out-of-sample behavior. Furthermore, learning a physically-consistent model of the underlying dynamics can subsequently enable usage of model-based controllers which can provide performance guarantees for complex, nonlinear systems. In particular, insight about

---

[*]This research was carried out during Y. D. Zhong's internship at Siemens Corporate Technology.

[1]We use the word *Symplectic* to emphasize that the learned dynamics endows a *symplectic* structure (Arnold et al., 2001) on the underlying space.

kinetic and potential energy of a physical system can be leveraged to synthesize appropriate control strategies, such as the method of controlled Lagrangian (Bloch et al., 2001) and interconnection & damping assignment (Ortega et al., 2002), which can reshape the closed-loop energy landscape to achieve a broad range of control objectives (regulation, tracking, etc.).

## RELATED WORK

**Physics-based Priors for Learning in Dynamical Systems:** The last few years have witnessed a significant interest in incorporating physics-based priors into deep learning frameworks. Such approaches, in contrast to more rigid parametric system identification techniques (Söderström & Stoica, 1988), use neural networks to approximate the state-transition dynamics and therefore are more expressive. Sanchez-Gonzalez et al. (2018), by representing the causal relationships in a physical system as a directed graph, use a recurrent graph network to infer latent space dynamics of robotic systems. Lutter et al. (2019) and Gupta et al. (2019) leverage Lagrangian mechanics to learn the dynamics of kinematic structures from time-series data of position, velocity, and acceleration. A more recent (concurrent) work by Greydanus et al. (2019) uses Hamiltonian mechanics to learn the dynamics of autonomous, energy-conserved mechanical systems from time-series data of position, momentum, and their derivatives. A key difference between these approaches and the proposed one is that our framework does not require any information about higher-order derivatives (e.g., acceleration) and can incorporate external control into the Hamiltonian formalism.

**Neural Networks for Dynamics and Control:** Inferring underlying dynamics from time-series data plays a critical role in controlling closed-loop response of dynamical systems, such as robotic manipulators (Lillicrap et al., 2015) and building HVAC systems (Wei et al., 2017). Although the use of neural networks towards identification and control of dynamical systems dates back to more than three decades ago (Narendra & Parthasarathy, 1990), recent advances in deep neural networks have led to renewed interest in this domain. Watter et al. (2015) learn dynamics with control from high-dimensional observations (raw image sequences) using a variational approach and synthesize an iterative LQR controller to control physical systems by imposing a locally linear constraint. Karl et al. (2016) and Krishnan et al. (2017) adopt a variational approach and use recurrent architectures to learn state-space models from noisy observation. SE3-Nets (Byravan & Fox, 2017) learn $SE(3)$ transformation of rigid bodies from point cloud data. Ayed et al. (2019) use partial information about the system state to learn a nonlinear state-space model. However, this body of work, while attempting to learn state-space models, does not take physics-based priors into consideration.

## CONTRIBUTION

The main contribution of this work is two-fold. First, we introduce a learning framework called Symplectic ODE-Net (SymODEN) which encodes a generalization of the Hamiltonian dynamics. This generalization, by adding an external control term to the standard Hamiltonian dynamics, allows us to learn the system dynamics which conforms to Hamiltonian dynamics with control. With the learned structured dynamics, we are able to synthesize controllers to control the system to track a reference configuration. Moreover, by encoding the structure, we can achieve better predictions with smaller network sizes. Second, we take one step forward in combining the physics-based prior and the data-driven approach. Previous approaches (Lutter et al., 2019; Greydanus et al., 2019) require data in the form of generalized coordinates and their derivatives up to the second order. However, a large number of physical systems accommodate generalized coordinates which are non-Euclidean (e.g., angles), and such angle data is often obtained in the embedded form, i.e., $(\cos q, \sin q)$ instead of the coordinate $(q)$ itself. The underlying reason is that an angular coordinate lies on $\mathbb{S}^1$ instead of $\mathbb{R}^1$. In contrast to previous approaches which do not address this aspect, SymODEN has been designed to work with angle data in the embedded form. Additionally, we leverage differentiable ODE solvers to avoid the need for estimating second-order derivatives of generalized coordinates. Code for the SymODEN framework and experiments is available at https://github.com/d-biswa/Symplectic-ODENet.

## 2 PRELIMINARY CONCEPTS

### 2.1 HAMILTONIAN DYNAMICS

Lagrangian dynamics and Hamiltonian dynamics are both reformulations of Newtonian dynamics. They provide novel insights into the laws of mechanics. In these formulations, the configuration of a system is described by its generalized coordinates. Over time, the configuration point of the system moves in the configuration space, tracing out a trajectory. Lagrangian dynamics describes the evolution of this trajectory, i.e., the equations of motion, in the configuration space. Hamiltonian dynamics, however, tracks the change of system states in the phase space, i.e. the product space of generalized coordinates $\mathbf{q} = (q_1, q_2, ..., q_n)$ and generalized momenta $\mathbf{p} = (p_1, p_2, ..., p_n)$. In other words, Hamiltonian dynamics treats $\mathbf{q}$ and $\mathbf{p}$ on an equal footing. This not only provides symmetric equations of motion but also leads to a whole new approach to classical mechanics (Goldstein et al., 2002). Hamiltonian dynamics is also widely used in statistical and quantum mechanics.

In Hamiltonian dynamics, the time-evolution of a system is described by the Hamiltonian $H(\mathbf{q}, \mathbf{p})$, a scalar function of generalized coordinates and momenta. Moreover, in almost all physical systems, the Hamiltonian is the same as the total energy and hence can be expressed as

$$H(\mathbf{q}, \mathbf{p}) = \frac{1}{2}\mathbf{p}^T\mathbf{M}^{-1}(\mathbf{q})\mathbf{p} + V(\mathbf{q}), \tag{1}$$

where the mass matrix $\mathbf{M}(\mathbf{q})$ is symmetric positive definite and $V(\mathbf{q})$ represents the potential energy of the system. Correspondingly, the time-evolution of the system is governed by

$$\dot{\mathbf{q}} = \frac{\partial H}{\partial \mathbf{p}} \qquad \dot{\mathbf{p}} = -\frac{\partial H}{\partial \mathbf{q}}, \tag{2}$$

where we have dropped explicit dependence on $\mathbf{q}$ and $\mathbf{p}$ for brevity of notation. Moreover, since

$$\dot{H} = \left(\frac{\partial H}{\partial \mathbf{q}}\right)^T \dot{\mathbf{q}} + \left(\frac{\partial H}{\partial \mathbf{p}}\right)^T \dot{\mathbf{p}} = 0, \tag{3}$$

the total energy is conserved along a trajectory of the system. The RHS of Equation (2) is called the symplectic gradient (Rowe et al., 1980) of $H$, and Equation (3) shows that moving along the symplectic gradient keeps the Hamiltonian constant.

In this work, we consider a generalization of the Hamiltonian dynamics which provides a means to incorporate external control ($\mathbf{u}$), such as force and torque. As external control is usually affine and only influences changes in the generalized momenta, we can express this generalization as

$$\begin{bmatrix} \dot{\mathbf{q}} \\ \dot{\mathbf{p}} \end{bmatrix} = \begin{bmatrix} \frac{\partial H}{\partial \mathbf{p}} \\ -\frac{\partial H}{\partial \mathbf{q}} \end{bmatrix} + \begin{bmatrix} \mathbf{0} \\ \mathbf{g}(\mathbf{q}) \end{bmatrix} \mathbf{u}, \tag{4}$$

where the input matrix $\mathbf{g}(\mathbf{q})$ is typically assumed to have full column rank. For $\mathbf{u} = \mathbf{0}$, the generalized dynamics reduces to the classical Hamiltonian dynamics (2) and the total energy is conserved; however, when $\mathbf{u} \neq \mathbf{0}$, the system has a dissipation-free energy exchange with the environment.

### 2.2 CONTROL VIA ENERGY SHAPING

Once we have learned the dynamics of a system, the learned model can be used to synthesize a controller for driving the system to a reference configuration $\mathbf{q}^\star$. As the proposed approach offers insight about the energy associated with a system, it is a natural choice to exploit this information for synthesizing controllers via *energy shaping* (Ortega et al., 2001). As energy is a fundamental aspect of physical systems, reshaping the associated energy landscape enables us to specify a broad range of control objectives and synthesize nonlinear controllers with provable performance guarantees.

If $\mathrm{rank}(\mathbf{g}(\mathbf{q})) = \mathrm{rank}(\mathbf{q})$, the system is fully-actuated and we have control over any dimension of "acceleration" in $\dot{\mathbf{p}}$. For such fully-actuated systems, a controller $\mathbf{u}(\mathbf{q}, \mathbf{p}) = \beta(\mathbf{q}) + \mathbf{v}(\mathbf{p})$ can be synthesized via *potential energy shaping* $\beta(\mathbf{q})$ and *damping injection* $\mathbf{v}(\mathbf{p})$. For completeness, we restate this procedure (Ortega et al., 2001) using our notation. As the name suggests, the goal of potential energy shaping is to synthesize $\beta(\mathbf{q})$ such that the closed-loop system behaves as if its time-evolution is governed by a desired Hamiltonian $H_d$. With this, we have

$$\begin{bmatrix} \dot{\mathbf{q}} \\ \dot{\mathbf{p}} \end{bmatrix} = \begin{bmatrix} \frac{\partial H}{\partial \mathbf{p}} \\ -\frac{\partial H}{\partial \mathbf{q}} \end{bmatrix} + \begin{bmatrix} \mathbf{0} \\ \mathbf{g}(\mathbf{q}) \end{bmatrix} \beta(\mathbf{q}) = \begin{bmatrix} \frac{\partial H_d}{\partial \mathbf{p}} \\ -\frac{\partial H_d}{\partial \mathbf{q}} \end{bmatrix}, \tag{5}$$

where the difference between the desired Hamiltonian and the original one lies in their potential energy term, i.e.

$$H_d(\mathbf{q}, \mathbf{p}) = \frac{1}{2}\mathbf{p}^T\mathbf{M}^{-1}(\mathbf{q})\mathbf{p} + V_d(\mathbf{q}). \tag{6}$$

In other words, $\boldsymbol{\beta}(\mathbf{q})$ *shape* the potential energy such that the desired Hamiltonian $H_d(\mathbf{q}, \mathbf{p})$ has a minimum at $(\mathbf{q}^\star, \mathbf{0})$. Then, by substituting Equation (1) and Equation (6) into Equation (5), we get

$$\boldsymbol{\beta}(\mathbf{q}) = \mathbf{g}^T(\mathbf{g}\mathbf{g}^T)^{-1}\Big(\frac{\partial V}{\partial \mathbf{q}} - \frac{\partial V_d}{\partial \mathbf{q}}\Big). \tag{7}$$

Thus, with potential energy shaping, we ensure that the system has the lowest energy at the desired reference configuration. Furthermore, to ensure that trajectories actually converge to this configuration, we add an additional damping term[2] given by

$$\mathbf{v}(\mathbf{p}) = -\mathbf{g}^T(\mathbf{g}\mathbf{g}^T)^{-1}(\mathbf{K}_d\mathbf{p}). \tag{8}$$

However, for underactuated systems, potential energy shaping alone cannot[3] drive the system to a desired configuration. We also need kinetic energy shaping for this purpose (Chang et al., 2002).

**Remark** If the desired potential energy is chosen to be a quadratic of the form

$$V_d(\mathbf{q}) = \frac{1}{2}(\mathbf{q} - \mathbf{q}^\star)^T\mathbf{K}_p(\mathbf{q} - \mathbf{q}^\star), \tag{9}$$

the external forcing term can be expressed as

$$\mathbf{u} = \mathbf{g}^T(\mathbf{g}\mathbf{g}^T)^{-1}\left(\frac{\partial V}{\partial \mathbf{q}} - \mathbf{K}_p(\mathbf{q} - \mathbf{q}^\star) - \mathbf{K}_d\mathbf{p}\right). \tag{10}$$

This can be interpreted as a PD controller with an additional energy compensation term. [4]

## 3   SYMPLECTIC ODE-NET

In this section, we introduce the network architecture of Symplectic ODE-Net. In Subsection 3.1, we show how to learn an ordinary differential equation with a constant control term. In Subsection 3.2, we assume we have access to generalized coordinate and momentum data and derive the network architecture. In Subsection 3.3, we take one step further to propose a data-driven approach to deal with data of embedded angle coordinates. In Subsection 3.4, we put together the line of reasoning introduced in the previous two subsections to propose SymODEN for learning dynamics on the hybrid space $\mathbb{R}^n \times \mathbb{T}^m$.

### 3.1   TRAINING NEURAL ODE WITH CONSTANT FORCING

Now we focus on the problem of learning the ordinary differential equation (ODE) from time series data. Consider an ODE: $\dot{\mathbf{x}} = \mathbf{f}(\mathbf{x})$. Assume we don't know the analytical expression of the right hand side (RHS) and we approximate it with a neural network. If we have time series data $\mathbf{X} = (\mathbf{x}_{t_0}, \mathbf{x}_{t_1}, ..., \mathbf{x}_{t_n})$, how could we learn $\mathbf{f}(\mathbf{x})$ from the data?

Chen et al. (2018) introduced Neural ODE, differentiable ODE solvers with O(1)-memory back-propagation. With Neural ODE, we make predictions by approximating the RHS function using a neural network $\mathbf{f}_\theta$ and feed it into an ODE solver

$$\hat{\mathbf{x}}_{t_1}, \hat{\mathbf{x}}_{t_2}, ..., \hat{\mathbf{x}}_{t_n} = \text{ODESolve}(\mathbf{x}_{t_0}, \mathbf{f}_\theta, t_1, t_2, ..., t_n)$$

We can then construct the loss function $L = \|\mathbf{X} - \hat{\mathbf{X}}\|_2^2$ and update the weights $\theta$ by backpropagating through the ODE solver.

In theory, we can learn $\mathbf{f}_\theta$ in this way. In practice, however, the neural net is hard to train if $n$ is large. If we have a bad initial estimate of the $\mathbf{f}_\theta$, the prediction error would in general be large. Although $|\mathbf{x}_{t_1} - \hat{\mathbf{x}}_{t_1}|$ might be small, $\hat{\mathbf{x}}_{t_N}$ would be far from $\mathbf{x}_{t_N}$ as error accumulates, which makes the neural network hard to train. In fact, the prediction error of $\hat{\mathbf{x}}_{t_N}$ is not as important as $\hat{\mathbf{x}}_{t_1}$. In other words, we should weight data points in a short time horizon more than the rest of the data points. In order

---

[2]If we have access to $\dot{\mathbf{q}}$ instead of $\mathbf{p}$, we use $\dot{\mathbf{q}}$ instead in Equation (8).

[3]As $gg^T$ is not invertible, we cannot solve the matching condition given by Equation (7).

[4]Please refer to Appendix B for more details.

to address this and better utilize the data, we introduce the time horizon $\tau$ as a hyperparameter and predict $\mathbf{x}_{t_{i+1}}, \mathbf{x}_{t_{i+2}}, ..., \mathbf{x}_{t_{i+\tau}}$ from initial condition $\mathbf{x}_{t_i}$, where $i = 0, ..., n - \tau$.

One challenge toward leveraging Neural ODE to learn state-space models is the incorporation of the control term into the dynamics. Equation (4) has the form $\dot{\mathbf{x}} = \mathbf{f}(\mathbf{x}, \mathbf{u})$ with $\mathbf{x} = (\mathbf{q}, \mathbf{p})$. A function of this form cannot be directly fed into Neural ODE directly since the domain and range of $\mathbf{f}$ have different dimensions. In general, if our data consist of trajectories of $(\mathbf{x}, \mathbf{u})_{t_0,...,t_n}$ where $\mathbf{u}$ remains the same in a trajectory, we can leverage the augmented dynamics

$$\begin{bmatrix} \dot{\mathbf{x}} \\ \dot{\mathbf{u}} \end{bmatrix} = \begin{bmatrix} \mathbf{f}_\theta(\mathbf{x}, \mathbf{u}) \\ \mathbf{0} \end{bmatrix} = \tilde{\mathbf{f}}_\theta(\mathbf{x}, \mathbf{u}). \tag{11}$$

With Equation (11), we can match the input and output dimension of $\tilde{\mathbf{f}}_\theta$, which enables us to feed it into Neural ODE. The idea here is to use different constant external forcing to get the system responses and use those responses to train the model. With a trained model, we can apply a time-varying $\mathbf{u}$ to the dynamics $\dot{\mathbf{x}} = \mathbf{f}_\theta(\mathbf{x}, \mathbf{u})$ and generate estimated trajectories. When we synthesize the controller, $\mathbf{u}$ remains constant in each integration step. As long as our model interpolates well among different values of constant $\mathbf{u}$, we could get good estimated trajectories with a time-varying $\mathbf{u}$. The problem is then how to design the network architecture of $\tilde{\mathbf{f}}_\theta$, or equivalently $\mathbf{f}_\theta$ such that we can learn the dynamics in an efficient way.

### 3.2 LEARNING FROM GENERALIZED COORDINATE AND MOMENTUM

Suppose we have trajectory data consisting of $(\mathbf{q}, \mathbf{p}, \mathbf{u})_{t_0,...,t_n}$, where $\mathbf{u}$ remains constant in a trajectory. If we have the prior knowledge that the unforced dynamics of $\mathbf{q}$ and $\mathbf{p}$ is governed by Hamiltonian dynamics, we can use three neural nets – $\mathbf{M}_{\theta_1}^{-1}(\mathbf{q})$, $V_{\theta_2}(\mathbf{q})$ and $\mathbf{g}_{\theta_3}(\mathbf{q})$ – as function approximators to represent the inverse of mass matrix, potential energy and the input matrix. Thus,

$$\mathbf{f}_\theta(\mathbf{q}, \mathbf{p}, \mathbf{u}) = \begin{bmatrix} \frac{\partial H_{\theta_1, \theta_2}}{\partial \mathbf{p}} \\ -\frac{\partial H_{\theta_1, \theta_2}}{\partial \mathbf{q}} \end{bmatrix} + \begin{bmatrix} \mathbf{0} \\ \mathbf{g}_{\theta_3}(\mathbf{q}) \end{bmatrix} \mathbf{u} \tag{12}$$

where

$$H_{\theta_1, \theta_2}(\mathbf{q}, \mathbf{p}) = \frac{1}{2} \mathbf{p}^T \mathbf{M}_{\theta_1}^{-1}(\mathbf{q}) \mathbf{p} + V_{\theta_2}(\mathbf{q}) \tag{13}$$

The partial derivative in the expression can be taken care of by automatic differentiation. by putting the designed $\mathbf{f}_\theta(\mathbf{q}, \mathbf{p}, \mathbf{u})$ into Neural ODE, we obtain a systematic way of adding the prior knowledge of Hamiltonian dynamics into end-to-end learning.

### 3.3 LEARNING FROM EMBEDDED ANGLE DATA

In the previous subsection, we assume $(\mathbf{q}, \mathbf{p}, \mathbf{u})_{t_0,...,t_n}$. In a lot of physical system models, the state variables involve angles which reside in the interval $[-\pi, \pi)$. In other words, each angle resides on the manifold $\mathbb{S}^1$. From a data-driven perspective, the data that respects the geometry is a 2 dimensional embedding $(\cos q, \sin q)$. Furthermore, the generalized momentum data is usually not available. Instead, the velocity is often available. For example, in OpenAI Gym (Brockman et al., 2016) Pendulum-v0 task, the observation is $(\cos q, \sin q, \dot{q})$.

From a theoretical perspective, however, the angle itself is often used, instead of the 2D embedding. The reason being both the Lagrangian and the Hamiltonian formulations are derived using generalized coordinates. Using an independent generalized coordinate system makes it easier to solve for the equations of motion.

In this subsection, we take the data-driven standpoint and develop an angle-aware method to accommodate the underlying manifold structure. We assume all the generalized coordinates are angles and the data comes in the form of $(\mathbf{x}_1(\mathbf{q}), \mathbf{x}_2(\mathbf{q}), \mathbf{x}_3(\dot{\mathbf{q}}), \mathbf{u})_{t_0,...,t_n} = (\cos \mathbf{q}, \sin \mathbf{q}, \dot{\mathbf{q}}, \mathbf{u})_{t_0,...,t_n}$. We aim to incorporate our theoretical prior – Hamiltonian dynamics – into the data-driven approach. The goal is to learn the dynamics of $\mathbf{x}_1$, $\mathbf{x}_2$ and $\mathbf{x}_3$. Noticing $\mathbf{p} = \mathbf{M}(\mathbf{x}_1, \mathbf{x}_2)\dot{\mathbf{q}}$, we can write down the derivative of $\mathbf{x}_1$, $\mathbf{x}_2$ and $\mathbf{x}_3$,

$$\begin{aligned} \dot{\mathbf{x}}_1 &= -\sin \mathbf{q} \circ \dot{\mathbf{q}} = -\mathbf{x}_2 \circ \dot{\mathbf{q}} \\ \dot{\mathbf{x}}_2 &= \cos \mathbf{q} \circ \dot{\mathbf{q}} = \mathbf{x}_1 \circ \dot{\mathbf{q}} \\ \dot{\mathbf{x}}_3 &= \frac{\mathrm{d}}{\mathrm{d}t}(\mathbf{M}^{-1}(\mathbf{x}_1, \mathbf{x}_2)\mathbf{p}) = \frac{\mathrm{d}}{\mathrm{d}t}(\mathbf{M}^{-1}(\mathbf{x}_1, \mathbf{x}_2))\mathbf{p} + \mathbf{M}^{-1}(\mathbf{x}_1, \mathbf{x}_2)\dot{\mathbf{p}} \end{aligned} \tag{14}$$

where "∘" represents the elementwise product (i.e., Hadamard product). We assume $\mathbf{q}$ and $\mathbf{p}$ evolve with the generalized Hamiltonian dynamics Equation (4). Here the Hamiltonian $H(\mathbf{x}_1, \mathbf{x}_2, \mathbf{p})$ is a function of $\mathbf{x}_1, \mathbf{x}_2$ and $\mathbf{p}$ instead of $\mathbf{q}$ and $\mathbf{p}$.

$$\dot{\mathbf{q}} = \frac{\partial H}{\partial \mathbf{p}} \tag{15}$$

$$\dot{\mathbf{p}} = -\frac{\partial H}{\partial \mathbf{q}} + \mathbf{g}(\mathbf{x}_1, \mathbf{x}_2)\mathbf{u} = -\frac{\partial \mathbf{x}_1}{\partial \mathbf{q}}\frac{\partial H}{\partial \mathbf{x}_1} - \frac{\partial \mathbf{x}_2}{\partial \mathbf{q}}\frac{\partial H}{\partial \mathbf{x}_2} + \mathbf{g}(\mathbf{x}_1, \mathbf{x}_2)\mathbf{u}$$

$$= \sin\mathbf{q} \circ \frac{\partial H}{\partial \mathbf{x}_1} - \cos\mathbf{q} \circ \frac{\partial H}{\partial \mathbf{x}_2} + \mathbf{g}(\mathbf{x}_1, \mathbf{x}_2)\mathbf{u} = \mathbf{x}_2 \circ \frac{\partial H}{\partial \mathbf{x}_1} - \mathbf{x}_1 \circ \frac{\partial H}{\partial \mathbf{x}_2} + \mathbf{g}(\mathbf{x}_1, \mathbf{x}_2)\mathbf{u} \tag{16}$$

Then the right hand side of Equation (14) can be expressed as a function of state variables and control $(\mathbf{x}_1, \mathbf{x}_2, \mathbf{x}_3, \mathbf{u})$. Thus, it can be fed into the Neural ODE. We use three neural nets – $\mathbf{M}_{\theta_1}^{-1}(\mathbf{x}_1, \mathbf{x}_2)$, $V_{\theta_2}(\mathbf{x}_1, \mathbf{x}_2)$ and $\mathbf{g}_{\theta_3}(\mathbf{x}_1, \mathbf{x}_2)$ – as function approximators. Substitute Equation (15) and Equation (16) into Equation (14), then the RHS serves as $\mathbf{f}_\theta(\mathbf{x}_1, \mathbf{x}_2, \mathbf{x}_3, \mathbf{u})$.[5]

$$\mathbf{f}_\theta(\mathbf{x}_1, \mathbf{x}_2, \mathbf{x}_3, \mathbf{u}) = \begin{bmatrix} -\mathbf{x}_2 \circ \frac{\partial H_{\theta_1, \theta_2}}{\partial \mathbf{p}} \\ \mathbf{x}_1 \circ \frac{\partial H_{\theta_1, \theta_2}}{\partial \mathbf{p}} \\ \frac{\mathrm{d}}{\mathrm{d}t}(\mathbf{M}_{\theta_1}^{-1}(\mathbf{x}_1, \mathbf{x}_2))\mathbf{p} + \mathbf{M}_{\theta_1}^{-1}(\mathbf{x}_1, \mathbf{x}_2)\left(\mathbf{x}_2 \circ \frac{\partial H_{\theta_1 \theta_2}}{\partial \mathbf{x}_1} - \mathbf{x}_1 \circ \frac{\partial H_{\theta_1 \theta_2}}{\partial \mathbf{x}_2} + \mathbf{g}_{\theta_3}(\mathbf{x}_1, \mathbf{x}_2)\mathbf{u}\right) \end{bmatrix} \tag{17}$$

where

$$H_{\theta_1, \theta_2}(\mathbf{x}_1, \mathbf{x}_2, \mathbf{p}) = \frac{1}{2}\mathbf{p}^T \mathbf{M}_{\theta_1}^{-1}(\mathbf{x}_1, \mathbf{x}_2)\mathbf{p} + V_{\theta_2}(\mathbf{x}_1, \mathbf{x}_2) \tag{18}$$

$$\mathbf{p} = \mathbf{M}_{\theta_1}(\mathbf{x}_1, \mathbf{x}_2)\mathbf{x}_3 \tag{19}$$

### 3.4 LEARNING ON HYBRID SPACES $\mathbb{R}^n \times \mathbb{T}^m$

In Subsection 3.2, we treated the generalized coordinates as translational coordinates. In Subsection 3.3, we developed an angle-aware method to better deal with embedded angle data. In most of physical systems, these two types of coordinates coexist. For example, robotics systems are usually modelled as interconnected rigid bodies. The positions of joints or center of mass are translational coordinates and the orientations of each rigid body are angular coordinates. In other words, the generalized coordinates lie on $\mathbb{R}^n \times \mathbb{T}^m$, where $\mathbb{T}^m$ denotes the $m$-torus, with $\mathbb{T}^1 = \mathbb{S}^1$ and $\mathbb{T}^2 = \mathbb{S}^1 \times \mathbb{S}^1$. In this subsection, we put together the architecture of the previous two subsections. We assume the generalized coordinates are $\mathbf{q} = (\mathbf{r}, \boldsymbol{\phi}) \in \mathbb{R}^n \times \mathbb{T}^m$ and the data comes in the form of $(\mathbf{x}_1, \mathbf{x}_2, \mathbf{x}_3, \mathbf{x}_4, \mathbf{x}_5, \mathbf{u})_{t_0, \ldots, t_n} = (\mathbf{r}, \cos\boldsymbol{\phi}, \sin\boldsymbol{\phi}, \dot{\mathbf{r}}, \dot{\boldsymbol{\phi}}, \mathbf{u})_{t_0, \ldots, t_n}$. With similar line of reasoning, we use three neural nets – $\mathbf{M}_{\theta_1}^{-1}(\mathbf{x}_1, \mathbf{x}_2, \mathbf{x}_3)$, $V_{\theta_2}(\mathbf{x}_1, \mathbf{x}_2, \mathbf{x}_3)$ and $\mathbf{g}_{\theta_3}(\mathbf{x}_1, \mathbf{x}_2, \mathbf{x}_3)$ – as function approximators. We have

$$\mathbf{p} = \mathbf{M}_{\theta_1}(\mathbf{x}_1, \mathbf{x}_2, , \mathbf{x}_3)\begin{bmatrix}\mathbf{x}_4 \\ \mathbf{x}_5\end{bmatrix} \tag{20}$$

$$H_{\theta_1, \theta_2}(\mathbf{x}_1, \mathbf{x}_2, \mathbf{x}_3, \mathbf{p}) = \frac{1}{2}\mathbf{p}^T \mathbf{M}_{\theta_1}^{-1}(\mathbf{x}_1, \mathbf{x}_2, \mathbf{x}_3)\mathbf{p} + V_{\theta_2}(\mathbf{x}_1, \mathbf{x}_2, \mathbf{x}_3) \tag{21}$$

with Hamiltonian dynamics, we have

$$\dot{\mathbf{q}} = \begin{bmatrix}\dot{\mathbf{r}} \\ \dot{\boldsymbol{\phi}}\end{bmatrix} = \frac{\partial H_{\theta_1, \theta_2}}{\partial \mathbf{p}} \tag{22}$$

$$\dot{\mathbf{p}} = \begin{bmatrix} -\frac{\partial H_{\theta_1, \theta_2}}{\partial \mathbf{x}_1} \\ \mathbf{x}_3 \circ \frac{\partial H_{\theta_1, \theta_2}}{\partial \mathbf{x}_2} - \mathbf{x}_2 \circ \frac{\partial H_{\theta_1, \theta_2}}{\partial \mathbf{x}_3} \end{bmatrix} + \mathbf{g}_{\theta_3}(\mathbf{x}_1, \mathbf{x}_2, \mathbf{x}_3)\mathbf{u} \tag{23}$$

Then

$$\begin{bmatrix}\dot{\mathbf{x}}_1 \\ \dot{\mathbf{x}}_2 \\ \dot{\mathbf{x}}_3 \\ \dot{\mathbf{x}}_4 \\ \dot{\mathbf{x}}_5\end{bmatrix} = \begin{bmatrix} \dot{\mathbf{r}} \\ -\mathbf{x}_3\dot{\boldsymbol{\phi}} \\ \mathbf{x}_2\dot{\boldsymbol{\phi}} \\ \frac{\mathrm{d}}{\mathrm{d}t}(\mathbf{M}_{\theta_1}^{-1}(\mathbf{x}_1, \mathbf{x}_2, \mathbf{x}_3))\mathbf{p} + \mathbf{M}_{\theta_1}^{-1}(\mathbf{x}_1, \mathbf{x}_2, \mathbf{x}_3)\dot{\mathbf{p}} \end{bmatrix} = \mathbf{f}_\theta(\mathbf{x}_1, \mathbf{x}_2, \mathbf{x}_3, \mathbf{x}_4, \mathbf{x}_5, \mathbf{u}) \tag{24}$$

where the $\dot{\mathbf{r}}$ and $\dot{\boldsymbol{\phi}}$ come from Equation (22). Now we obtain a $\mathbf{f}_\theta$ which can be fed into Neural ODE. Figure 1 shows the flow of the computation graph based on Equation (20)-(24).

---

[5]In Equation (17), the derivative of $\mathbf{M}_{\theta_1}^{-1}(\mathbf{x}_1, \mathbf{x}_2)$ can be expanded using chain rule and expressed as a function of the states.

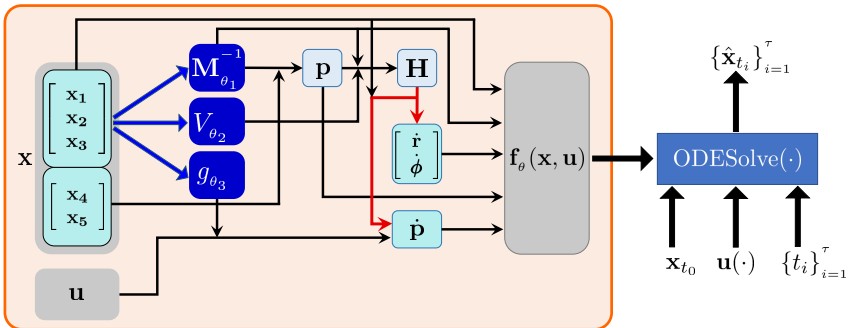

Figure 1: The computation graph of SymODEN. Blue arrows indicate neural network parametrization. Red arrows indicate automatic differentiation. For a given $(\mathbf{x}, \mathbf{u})$, the computation graph outputs a $\mathbf{f}_\theta(\mathbf{x}, \mathbf{u})$ which follows Hamiltonian dynamics with control. The function itself is an input to the Neural ODE to generate estimation of states at each time step. Since all the operations are differentiable, weights of the neural networks can be updated by backpropagation.

### 3.5 POSITIVE DEFINITENESS OF THE MASS MATRIX

In real physical systems, the mass matrix $\mathbf{M}$ is positive definite, which ensures a positive kinetic energy with a non-zero velocity. The positive definiteness of $\mathbf{M}$ implies the positive definiteness of $\mathbf{M}_{\theta_1}^{-1}$. Thus, we impose this constraint in the network architecture by $\mathbf{M}_{\theta_1}^{-1} = \mathbf{L}_{\theta_1}\mathbf{L}_{\theta_1}^T$, where $\mathbf{L}_{\theta_1}$ is a lower-triangular matrix. The positive definiteness is ensured if the diagonal elements of $\mathbf{M}_{\theta_1}^{-1}$ are positive. In practice, this can be done by adding a small constant $\epsilon$ to the diagonal elements of $\mathbf{M}_{\theta_1}^{-1}$. It not only makes $\mathbf{M}_{\theta_1}$ invertible, but also stabilize the training.

## 4 EXPERIMENTS

### 4.1 EXPERIMENTAL SETUP

We use the following four tasks to evaluate the performance of Symplectic ODE-Net model - (i) Task 1: a pendulum with generalized coordinate and momentum data (learning on $\mathbb{R}^1$); (ii) Task 2: a pendulum with embedded angle data (learning on $\mathbb{S}^1$); (iii) Task 3: a CartPole system (learning on $\mathbb{R}^1 \times \mathbb{S}^1$); and (iv) Task 4: an Acrobot (learning on $\mathbb{T}^2$).

**Model Variants**. Besides the Symplectic ODE-Net model derived above, we consider a variant by approximating the Hamiltonian using a fully connected neural net $H_{\theta_1, \theta_2}$. We call it *Unstructured Symplectic ODE-Net (Unstructured SymODEN)* since this model does not exploit the structure of the Hamiltonian (1).

**Baseline Models**. In order to show that we can learn the dynamics better with less parameters by leveraging prior knowledge, we set up baseline models for all four experiments. For the pendulum with generalized coordinate and momentum data, the *naive baseline* model approximates Equation (12) – $\mathbf{f}_\theta(\mathbf{x}, \mathbf{u})$ – by a fully connected neural net. For all the other experiments, which involves embedded angle data, we set up two different baseline models: *naive baseline* approximates $\mathbf{f}_\theta(\mathbf{x}, \mathbf{u})$ by a fully connected neural net. It doesn't respect the fact that the coordinate pair, $\cos\boldsymbol{\phi}$ and $\sin\boldsymbol{\phi}$, lie on $\mathbb{T}^m$. Thus, we set up the *geometric baseline* model which approximates $\dot{q}$ and $\dot{p}$ with a fully connected neural net. This ensures that the angle data evolves on $\mathbb{T}^m$. [6]

**Data Generation**. For all tasks, we randomly generated initial conditions of states and subsequently combined them with 5 different constant control inputs, i.e., $u = -2.0, -1.0, 0.0, 1.0, 2.0$ to produce the initial conditions and input required for simulation. The simulators integrate the corresponding dynamics for 20 time steps to generate trajectory data which is then used to construct the training set. The simulators for different tasks are different. For Task 1, we integrate the true generalized Hamiltonian dynamics with a time interval of 0.05 seconds to generate trajectories. All the other tasks deal with embedded angle data and velocity directly, so we use OpenAI Gym (Brockman et al., 2016) simulators to generate trajectory data. One drawback of using OpenAI Gym is that not all environments use the Runge-Kutta method (RK4) to carry out the integration. OpenAI Gym favors other numerical schemes over RK4 because of speed, but it is harder to learn the dynamics with

---

[6]For more information on model details, please refer to Appendix A.

inaccurate data. For example, if we plot the total energy as a function of time from data generated by `Pendulum-v0` environment with zero action, we see that the total energy oscillates around a constant by a significant amount, even though the total energy should be conserved. Thus, for Task 2 and Task 3, we use `Pendulum-v0` and `CartPole-v1`, respectively, and replace the numerical integrator of the environments to RK4. For Task 4, we use the `Acrobot-v1` environment which is already using RK4. We also change the action space of `Pendulum-v0`, `CartPole-v1` and `Acrobot-v1` to a continuous space with a large enough bound.

**Model training**. In all the tasks, we train our model using Adam optimizer (Kingma & Ba, 2014) with 1000 epochs. We set a time horizon $\tau = 3$, and choose "RK4" as the numerical integration scheme in Neural ODE. We vary the size of the training set by doubling from 16 initial state conditions to 1024 initial state conditions. Each initial state condition is combined with five constant control $u = -2.0, -1.0, 0.0, 1.0, 2.0$ to produce initial condition for simulation. Each trajectory is generated by integrating the dynamics 20 time steps forward. We set the size of mini-batches to be the number of initial state conditions. We logged the *train error* per trajectory and the *prediction error* per trajectory in each case for all the tasks. The train error per trajectory is the mean squared error (MSE) between the estimated trajectory and the ground truth over 20 time steps. To evaluate the performance of each model in terms of long time prediction, we construct the metric of prediction error per trajectory by using the same initial state condition in the training set with a constant control of $u = 0.0$, integrating 40 time steps forward, and calculating the MSE over 40 time steps. The reason for using only the unforced trajectories is that a constant nonzero control might cause the velocity to keep increasing or decreasing over time, and large absolute values of velocity are of little interest for synthesizing controllers.

### 4.2 TASK 1: PENDULUM WITH GENERALIZED COORDINATE AND MOMENTUM DATA

In this task, we use the model described in Section 3.2 and present the predicted trajectories of the learned models as well as the learned functions of SymODEN. We also point out the drawback of treating the angle data as a Cartesian coordinate. The dynamics of this task has the following form

$$\dot{q} = 3p, \qquad \dot{p} = -5\sin q + u \quad (25)$$

with Hamiltonian $H(q, p) = 1.5p^2 + 5(1 - \cos q)$. In other words $M(q) = 3$, $V(q) = 5(1 - \cos q)$ and $g(q) = 1$.

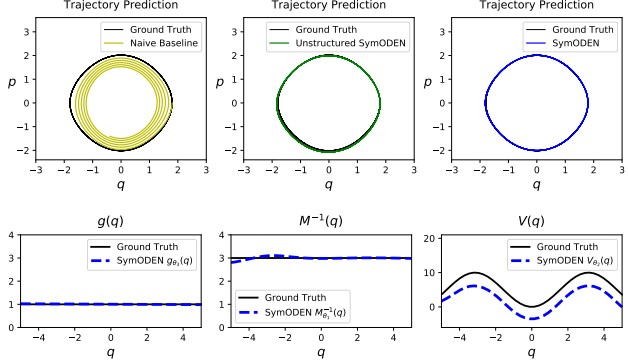

Figure 2: Sample trajectories and learned functions of Task 1.

In Figure 2, The ground truth is an unforced trajectory which is energy-conserved. The prediction trajectory of the baseline model does not conserve energy, while both the SymODEN and its unstructured variant predict energy-conserved trajectories. For SymODEN, the learned $g_{\theta_3}(q)$ and $M_{\theta_1}^{-1}(q)$ matches the ground truth well. $V_{\theta_2}(q)$ differs from the ground truth with a constant. This is acceptable since the potential energy is a relative notion. Only the derivative of $V_{\theta_2}(q)$ plays a role in the dynamics.

Here we treat $q$ as a variable in $\mathbb{R}^1$ and our training set contains initial conditions of $q \in [-\pi, 3\pi]$. The learned functions do not extrapolate well outside this range, as we can see from the left part in the figures of $M_{\theta_1}^{-1}(q)$ and $V_{\theta_2}(q)$. We address this issue by working directly with embedded angle data, which leads us to the next subsection.

### 4.3 TASK 2: PENDULUM WITH EMBEDDED DATA

In this task, the dynamics is the same as Equation (25) but the training data are generated by the OpenAI Gym simulator, i.e. we use embedded angle data and assume we only have access to $\dot{q}$ instead of $p$. We use the model described in Section 3.3 and synthesize an energy-based controller (Section 2.2). Without true $p$ data, the learned function matches the ground truth with a scaling $\beta$, as shown in Figure 3. To explain the scaling, let us look at the following dynamics

$$\dot{q} = p/\alpha, \qquad \dot{p} = -15\alpha \sin q + 3\alpha u \qquad (26)$$

with Hamiltonian $H = p^2/(2\alpha) + 15\alpha(1 - \cos q)$. If we only look at the dynamics of $q$, we have $\ddot{q} = -15\sin q + 3u$, which is independent of $\alpha$. If we don't have access to the generalized momentum $p$, our trained neural network may converge to a Hamiltonian with a $\alpha_e$ which is

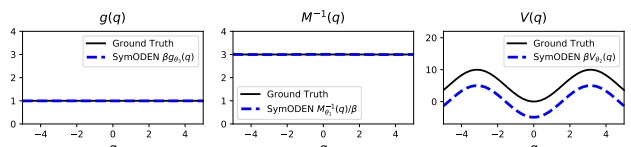

Figure 3: Without true generalized momentum data, the learned functions match the ground truth with a scaling. Here $\beta = 0.357$

different from the true value, $\alpha_t = 1/3$, in this task. By a scaling $\beta = \alpha_t/\alpha_e = 0.357$, the learned functions match the ground truth. Even we are not learning the true $\alpha_t$, we can still perform prediction and control since we are learning the dynamics of $q$ correctly. We let $V_d = -V_{\theta_2}(q)$, then the desired Hamiltonian has minimum energy when the pendulum rests at the upward position. For the damping injection, we let $K_d = 3$. Then from Equation (7) and (8), the controller we synthesize is

$$u(\cos q, \sin q, \dot{q}) = g_{\theta_3}^{-1}(\cos q, \sin q)\Big(2\big(-\frac{\partial V_{\theta_2}}{\partial \cos q}\sin q + \frac{\partial V_{\theta_2}}{\partial \sin q}\cos q\big) - 3\dot{q}\Big) \qquad (27)$$

Only SymODEN out of all models we consider provides the learned potential energy which is required to synthesize the controller. Figure 4 shows how the states evolve when the controller is fed into the OpenAI Gym simulator. We can successfully

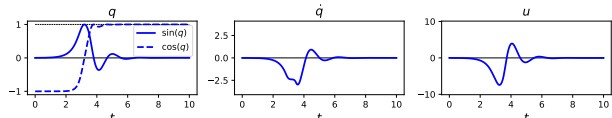

Figure 4: Time-evolution of the state variables $(\cos q, \sin q, \dot{q})$ when the closed-loop control input $u(\cos q, \sin q, \dot{q})$ is governed by Equation (27). The thin black lines show the expected results.

control the pendulum into the inverted position using the controller based on the learned model even though the absolute maximum control $u$, 7.5, is more than three times larger than the absolute maximum $u$ in the training set, which is 2.0. This shows SymODEN extrapolates well.

### 4.4 TASK 3: CARTPOLE SYSTEM

The CartPole system is an underactuated system and to synthesize a controller to balance the pole from arbitrary initial condition requires trajectory optimization or kinetic energy shaping. We show that we can learn its dynamics and perform prediction in Section 4.6. We also train SymODEN in a fully-actuated version of the CartPole system (see Appendix E). The corresponding energy-based controller can bring the pole to the inverted position while driving the cart to the origin.

### 4.5 TASK 4: ACROBOT

The Acrobot is an underactuated double pendulum. As this system exhibits chaotic motion, it is not possible to predict its long-term behavior. However, Figure 6 shows that SymODEN can provide reasonably good short-term prediction. We also train SymODEN in a fully-actuated version of the Acrobot and show that we can control this system to reach the inverted position (see Appendix E).

### 4.6 RESULTS

In this subsection, we show the train error, prediction error, as well as the MSE and total energy of a sample test trajectory for all the tasks. Figure 5 shows the variation in train error and prediction error with changes in the number of initial state conditions in the training set. We can see that SymODEN yields better generalization in every task. In Task 3, although the Geometric Baseline Model yields lower train error in comparison to the other models, SymODEN generates more accurate predictions, indicating overfitting in the Geometric Baseline Model. By incorporating the physics-based prior of Hamiltonian dynamics, SymODEN learns dynamics that obeys physical laws and thus provides better predictions. In most cases, SymODEN trained with a smaller training dataset performs better than other models in terms of the train and prediction error, indicating that better generalization can be achieved even with fewer training samples.

Figure 6 shows the evolution of MSE and total energy along a trajectory with a previously unseen initial condition. For all the tasks, MSE of the baseline models diverges faster than SymODEN. Unstructured SymODEN performs well in all tasks except Task 3. As for the total energy, in Task 1 and Task 2, SymODEN and Unstructured SymODEN conserve total energy by oscillating around a constant value. In these models, the Hamiltonian itself is learned and the prediction of the future states

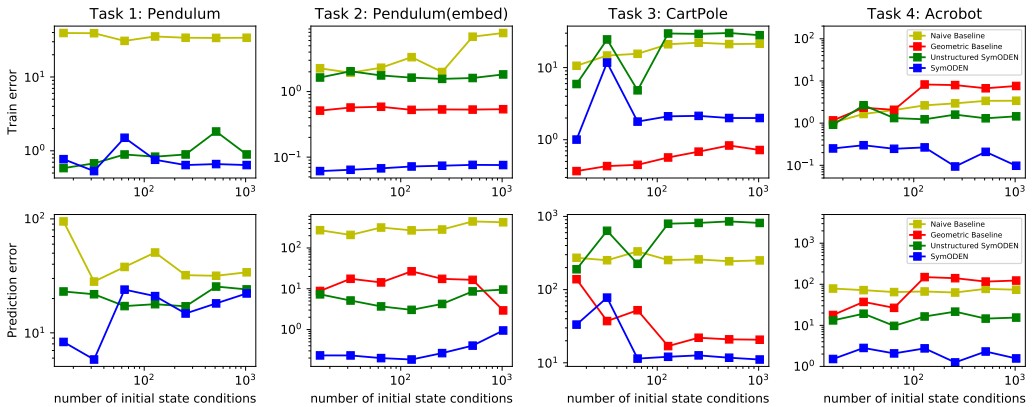

Figure 5: Train error per trajectory and prediction error per trajectory for all 4 tasks with different number of training trajectories. Horizontal axis shows number of initial state conditions (16, 32, 64, 128, 256, 512, 1024) in the training set. Both the horizontal axis and vertical axis are in log scale.

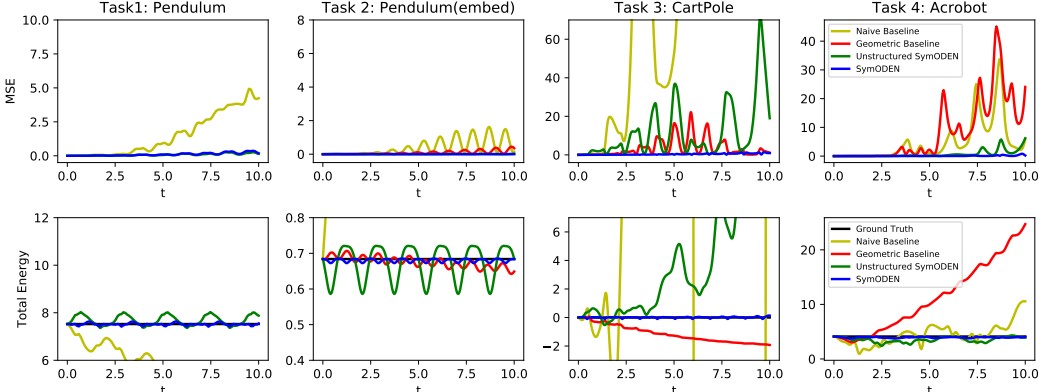

Figure 6: Mean square error and total energy of test trajectories. SymODEN works the best in terms of both MSE and total energy. Since SymODEN has learned the Hamiltonian and discovered the conservation from data the predicted trajectories match the ground truth. The ground truth of energy in all four tasks stay constant.

stay around a level set of the Hamiltonian. Baseline models, however, fail to find the conservation and the estimation of future states drift away from the initial Hamiltonian level set.

## 5 CONCLUSION

Here we have introduced Symplectic ODE-Net which provides a systematic way to incorporate prior knowledge of Hamiltonian dynamics with control into a deep learning framework. We show that SymODEN achieves better prediction with fewer training samples by learning an interpretable, physically-consistent state-space model. Future works will incorporate a broader class of physics-based prior, such as the port-Hamiltonian system formulation, to learn dynamics of a larger class of physical systems. SymODEN can work with embedded angle data or when we only have access to velocity instead of generalized momentum. Future works would explore other types of embedding, such as embedded 3D orientations. Another interesting direction could be to combine energy shaping control (potential as well as kinetic energy shaping) with interpretable end-to-end learning frameworks.

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

# Appendices

## A   EXPERIMENT IMPLEMENTATION DETAILS

The architectures used for our experiments are shown below. For all the tasks, SymODEN has the lowest number of total parameters. To ensure that the learned function is smooth, we use Tanh activation function instead of ReLu. As we have differentiation in the computation graph, non-smooth activation functions would lead to discontinuities in the derivatives. This, in turn, would result in an ODE with a discontinuous RHS which is not desirable. All the architectures shown below are fully-connected neural networks. The first number indicates the dimension of the input layer. The last number indicates the dimension of output layer. The dimension of hidden layers is shown in the middle along with the activation functions.

**Task 1: Pendulum**

- Input: 2 state dimensions, 1 action dimension
- Baseline Model (0.36M parameters): 2 - 600Tanh - 600Tanh - 2Linear
- Unstructured SymODEN (0.20M parameters):
    - $H_{\theta_1, \theta_2}$: 2 - 400Tanh - 400Tanh - 1Linear
    - $g_{\theta_3}$: 1 - 200Tanh - 200Tanh - 1Linear
- SymODEN (0.13M parameters):
    - $M_{\theta_1}^{-1}$: 1 - 300Tanh - 300Tanh - 1Linear
    - $V_{\theta_2}$: 1 - 50Tanh - 50Tanh - 1Linear
    - $g_{\theta_3}$: 1 - 200Tanh - 200Tanh - 1Linear

**Task 2: Pendulum with embedded data**

- Input: 3 state dimensions, 1 action dimension

- Naive Baseline Model (0.65M parameters): 4 - 800Tanh - 800Tanh - 3Linear
- Geometric Baseline Model (0.46M parameters):
    - $M_{\theta_1}^{-1} = L_{\theta_1} L_{\theta_1}^T$, where $L_{\theta_1}$: 2 - 300Tanh - 300Tanh - 300Tanh - 1Linear
    - approximate $(\dot{q}, \dot{p})$: 4 - 600Tanh - 600Tanh - 2Linear
- Unstructured SymODEN (0.39M parameters):
    - $M_{\theta_1}^{-1} = L_{\theta_1} L_{\theta_1}^T$, where $L_{\theta_1}$: 2 - 300Tanh - 300Tanh - 300Tanh - 1Linear
    - $H_{\theta_2}$: 3 - 500Tanh - 500Tanh - 1Linear
    - $g_{\theta_3}$: 2 - 200Tanh - 200Tanh - 1Linear
- SymODEN (0.14M parameters):
    - $M_{\theta_1}^{-1} = L_{\theta_1} L_{\theta_1}^T$, where $L_{\theta_1}$: 2 - 300Tanh - 300Tanh - 300Tanh - 1Linear
    - $V_{\theta_2}$: 2 - 50Tanh - 50Tanh - 1Linear
    - $g_{\theta_3}$: 2 - 200Tanh - 200Tanh - 1Linear

### Task 3: CartPole

- Input: 5 state dimensions, 1 action dimension
- Naive Baseline Model (1.01M parameters): 6 - 1000Tanh - 1000Tanh - 5Linear
- Geometric Baseline Model (0.82M parameters):
    - $M_{\theta_1}^{-1} = L_{\theta_1} L_{\theta_1}^T$, where $L_{\theta_1}$: 3 - 400Tanh - 400Tanh - 400Tanh - 3Linear
    - approximate $(\dot{\mathbf{q}}, \dot{\mathbf{p}})$: 6 - 700Tanh - 700Tanh - 4Linear
- Unstructured SymODEN (0.67M parameters):
    - $M_{\theta_1}^{-1} = L_{\theta_1} L_{\theta_1}^T$, where $L_{\theta_1}$: 3 - 400Tanh - 400Tanh - 400Tanh - 3Linear
    - $H_{\theta_2}$: 5 - 500Tanh - 500Tanh - 1Linear
    - $g_{\theta_3}$: 3 - 300Tanh - 300Tanh - 2Linear
- SymODEN (0.51M parameters):
    - $M_{\theta_1}^{-1} = L_{\theta_1} L_{\theta_1}^T$, where $L_{\theta_1}$: 3 - 400Tanh - 400Tanh - 400Tanh - 3Linear
    - $V_{\theta_2}$: 3 - 300Tanh - 300Tanh - 1Linear
    - $g_{\theta_3}$: 3 - 300Tanh - 300Tanh - 2Linear

### Task 4: Acrobot

- Input: 6 state dimensions, 1 action dimension
- Naive Baseline Model (1.46M parameters): 7 - 1200Tanh - 1200Tanh - 6Linear
- Geometric Baseline Model (0.97M parameters):
    - $M_{\theta_1}^{-1} = L_{\theta_1} L_{\theta_1}^T$, where $L_{\theta_1}$: 4 - 400Tanh - 400Tanh - 400Tanh - 3Linear
    - approximate $(\dot{\mathbf{q}}, \dot{\mathbf{p}})$: 7 - 800Tanh - 800Tanh - 4Linear
- Unstructured SymODEN (0.78M parameters):
    - $M_{\theta_1}^{-1} = L_{\theta_1} L_{\theta_1}^T$, where $L_{\theta_1}$: 4 - 400Tanh - 400Tanh - 400Tanh - 3Linear
    - $H_{\theta_2}$: 6 - 600Tanh - 600Tanh - 1Linear
    - $g_{\theta_3}$: 4 - 300Tanh - 300Tanh - 2Linear
- SymODEN (0.51M parameters):
    - $M_{\theta_1}^{-1} = L_{\theta_1} L_{\theta_1}^T$, where $L_{\theta_1}$: 4 - 400Tanh - 400Tanh - 400Tanh - 3Linear
    - $V_{\theta_2}$: 4 - 300Tanh - 300Tanh - 1Linear
    - $g_{\theta_3}$: 4 - 300Tanh - 300Tanh - 2Linear

# B  SPECIAL CASE OF ENERGY-BASED CONTROLLER - PD CONTROLLER WITH ENERGY COMPENSATION

The energy-based controller has the form $\mathbf{u}(\mathbf{q}, \mathbf{p}) = \boldsymbol{\beta}(\mathbf{q}) + \mathbf{v}(\mathbf{p})$, where the potential energy shaping term $\boldsymbol{\beta}(\mathbf{q})$ and the damping injection term $\mathbf{v}(\mathbf{p})$ are given by Equation (7) and Equation (8), respectively.

If the desired potential energy $V_q(\mathbf{q})$ is given by a quadratic, as in Equation (9), then

$$
\begin{aligned}
\boldsymbol{\beta}(\mathbf{q}) &= \mathbf{g}^T (\mathbf{g}\mathbf{g}^T)^{-1} \Big( \frac{\partial V}{\partial \mathbf{q}} - \frac{\partial V_d}{\partial \mathbf{q}} \Big) \\
&= \mathbf{g}^T (\mathbf{g}\mathbf{g}^T)^{-1} \Big( \frac{\partial V}{\partial \mathbf{q}} - \mathbf{K}_p(\mathbf{q} - \mathbf{q}^\star) \Big),
\end{aligned}
\tag{28}
$$

and the controller can be expressed as

$$
\mathbf{u}(\mathbf{q}, \mathbf{p}) = \boldsymbol{\beta}(\mathbf{q}) + \mathbf{v}(\mathbf{p}) = \mathbf{g}^T (\mathbf{g}\mathbf{g}^T)^{-1} \Big( \frac{\partial V}{\partial \mathbf{q}} - \mathbf{K}_p(\mathbf{q} - \mathbf{q}^\star) - \mathbf{K}_d \mathbf{p} \Big).
\tag{29}
$$

The corresponding external forcing term is then given by

$$
\mathbf{g}(\mathbf{q})\mathbf{u} = \frac{\partial V}{\partial \mathbf{q}} - \mathbf{K}_p(\mathbf{q} - \mathbf{q}^\star) - \mathbf{K}_d \mathbf{p},
\tag{30}
$$

which is same as Equation (10) in the main body of the paper. The first term in this external forcing provides an energy compensation, whereas the second term and the last term are proportional and derivative control terms, respectively. Thus, this control can be perceived as a PD controller with an additional energy compensation.

# C  ABLATION STUDY OF DIFFERENTIABLE ODE SOLVER

In Hamiltonian Neural Networks (HNN), Greydanus et al. (2019) incorporate the Hamiltonian structure into learning by minimizing the difference between the symplectic gradients and the true gradients. When the true gradient is not available, which is often the case, the authors suggested using finite difference approximations. In SymODEN, true gradients or gradient approximations are not necessary since we integrate the estimated gradient using differentiable ODE solvers and set up the loss function with the integrated values. Here we perform an ablation study of the differentiable ODE Solver.

Both HNN and the Unstructured SymODEN approximate the Hamiltonian by a neural network and the main difference is the differentiable ODE solver, so we compare the performance of HNN and the Unstructured SymODEN. We set the time horizon $\tau = 1$ since it naturally corresponds to the finite difference estimate of the gradient. A larger $\tau$ would correspond to higher-order estimates of gradients. Since there is no angle-aware design in HNN, we use Task 1 to compare the performance of these two models.

We generate 25 training trajectories, each of which contains 45 time steps. This is consistent with the HNN paper. In the HNN paper Greydanus et al. (2019), the initial conditions of the trajectories are generated randomly in an annulus, whereas in this paper, we generate the initial state conditions uniformly in a reasonable range in each state dimension. We guess the reason that the authors of HNN choose the annulus data generation is that they do not have an angle-aware design. Take the pendulum for example; all the training and test trajectories they generate do not pass the inverted position. If they make prediction on a trajectory with a large enough initial speed, the angle would go over $\pm 2\pi$, $\pm 4\pi$, etc. in the long run. Since these are away from the region where the model gets trained, we can expect the prediction would be poor. In fact, this motivates us to design the angle-aware SymODEN in Section 3.3. In this ablation study, we generate the training data in both ways.

Table 1 shows the train error and the prediction error per trajectory of the two models. We can see Unstructured SymODEN performs better than HNN. This is an expected result. To see why this is the case, let us assume the training loss per time step of HNN is similar to that of Unstructured SymODEN. Since the training loss is on the symplectic gradient, the error would accumulate while integrating the symplectic gradient to get the estimated state values, and MSE of the state values

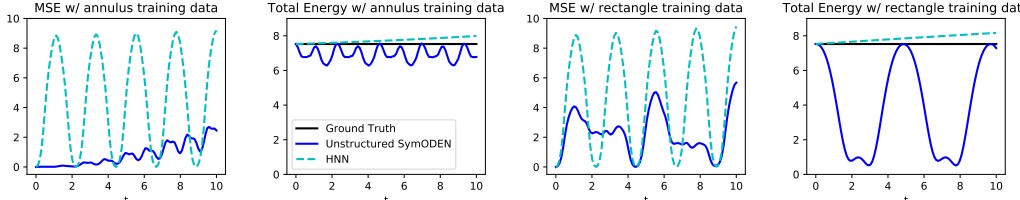

Figure 7: MSE and Total energy of a sample test trajectory. Left two figures: the training data for the models are randomly generated in an annulus, the same as in HNN. Right two figures: the training data for the models are randomly generated in a rectangle - the same way that we use in SymODEN.

would likely be one order of magnitude greater than that of Unstructured SymODEN. Figure 7 shows the MSE and total energy of a particular trajectory. It is clear that the MSE of the Unstructured SymODEN is lower than that of HNN. The MSE of HNN periodically touches zero does not mean it has a good prediction at that time step. Since the trajectories in the phase space are closed circles, those zeros mean the predicted trajectory of HNN lags behind (or runs ahead of) the true trajectory by one or more circles. Also, the energy of the HNN trajectory drifts instead of staying constant, probably because the finite difference approximation is not accurate enough.

Table 1: Train error and prediction error per trajectory of Unstructured SymODEN and HNN. The train error per trajectory is the sum of MSE of all the 45 timesteps averaged over the 25 training trajectories. The prediction error per trajectory is the sum of MSE of 90 timesteps in a trajectory.

| Models | annulus training data | | rectangle training data | |
|---|---|---|---|---|
| | train error | prediction error | train error | prediction error |
| Unstructured SymODEN | 56.59 | 440.78 | 502.60 | 4363.87 |
| HNN | 290.67 | 564.16 | 5457.80 | 26209.17 |

## D    EFFECTS OF THE TIME HORIZON $\tau$

Incorporating the differential ODE solver also introduces two hyperparameters: solver types and time horizon $\tau$. For the solver types, the Euler solver is not accurate enough for our tasks. The adaptive solver "dopri5" lead to similar train error, test error and prediction error as the RK4 solver, but requires more time during training. Thus, in our experiments, we choose RK4.

Time horizon $\tau$ is the number of points we use to construct our loss function. Table 2 shows the train error, test error and prediction error per trajectory in Task 2 when $\tau$ is varied from 1 to 5. We can see that longer time horizons lead to better models. This is expected since long time horizons penalize worse long term predictions. We also observe in our experiments that longer time horizons require more time to train the models.

Table 2: Train error, test error and prediction error per trajectory of Task 2

| Time Horizon | $\tau = 1$ | $\tau = 2$ | $\tau = 3$ | $\tau = 4$ | $\tau = 5$ |
|---|---|---|---|---|---|
| Train Error | 0.744 | 0.136 | 0.068 | 0.033 | 0.017 |
| Test Error | 0.579 | 0.098 | 0.052 | 0.024 | 0.012 |
| Prediction Error | 3.138 | 0.502 | 0.199 | 0.095 | 0.048 |

## E    FULLY-ACTUATED CARTPOLE AND ACROBOT

CartPole and Acrobot are underactuated systems. Incorporating the control of underactuated systems into the end-to-end learning framework is our future work. Here we trained SymODEN on fully-actuated versions of Cartpole and Acrobot and synthesized controllers based on the learned model.

For the fully-actuated CartPole, Figure 8 shows the snapshots of the system of a controlled trajectory with an initial condition where the pole is below the horizon. Figure 9 shows the time series of state

variables and control inputs. We can successfully learn the dynamics and control the pole to the inverted position and the cart to the origin.

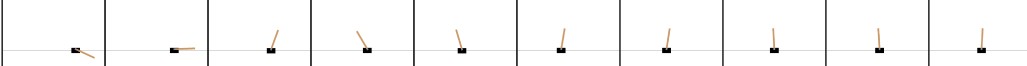

Figure 8: Snapshots of a controlled trajectory of the fully-actuated CartPole system with a 0.3s time interval.

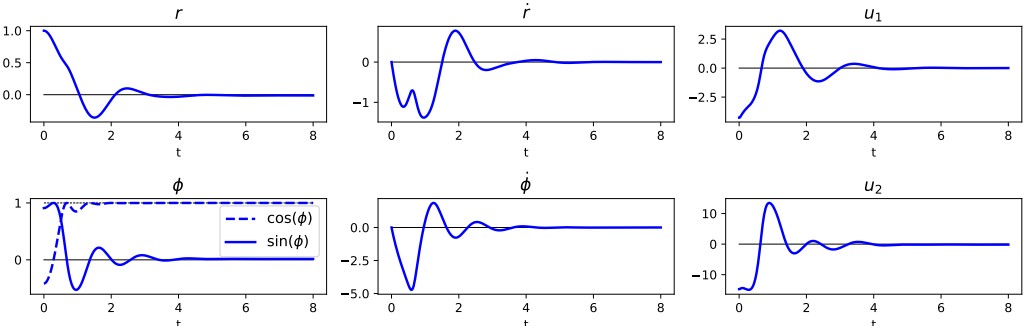

Figure 9: Time series of state variables and control inputs of a controlled trajectory shown in Figure 8. Black reference lines indicate expected value in the end.

For the fully-actuated Acrobot, Figure 10 shows the snapshots of a controlled trajectory. Figure 11 shows the time series of state variables and control inputs. We can successfully control the Acrobot from the downward position to the upward position, though the final value of $q_2$ is a little away from zero. Taking into account that the dynamics has been learned with only 64 different initial state conditions, it is most likely that the upward position did not show up in the training data.

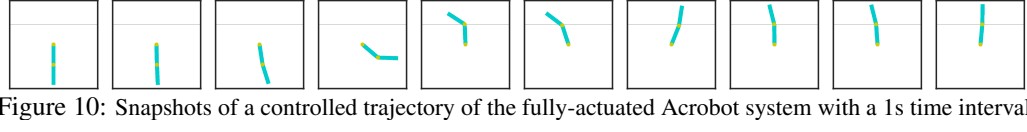

Figure 10: Snapshots of a controlled trajectory of the fully-actuated Acrobot system with a 1s time interval.

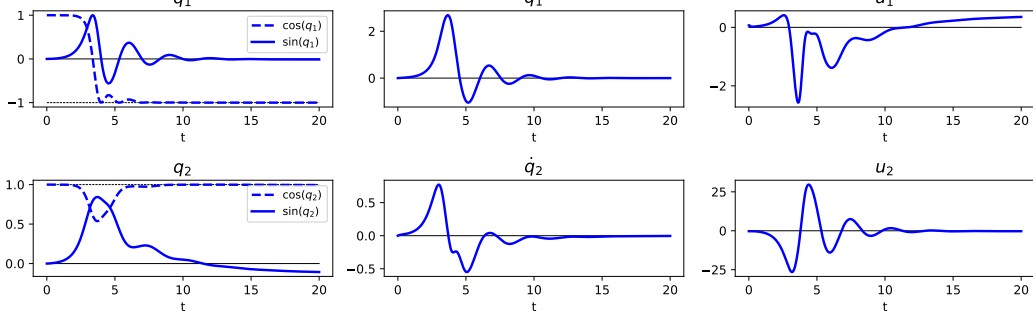

Figure 11: Time series of state variables and control inputs of a controlled trajectory shown in Figure 10. Black reference lines indicate expected value in the end.

## F    TEST ERRORS OF THE TASKS

Here we show statistics of train, test, and prediction per trajectory in all four tasks. The train errors are based on 64 initial state conditions and 5 constant inputs. The test errors are based on 64 previously unseen initial state conditions and the same 5 constant inputs. Each trajectory in the train and test set contains 20 steps. The prediction error is based on the same 64 initial state conditions (during training) and zero inputs.

Table 3: Train, Test and Prediction errors of the Four Tasks

|  | Naive Baseline | Geometric Baseline | Unstructured Symplectic-ODE | Symplectic-ODE |
|---|---|---|---|---|
| **Task 1: Pendulum** | | | | |
| Model Parameter | 0.36M | N/A | 0.20M | 0.13M |
| Train error | $30.82 \pm 43.45$ | N/A | $0.89 \pm 2.76$ | $1.50 \pm 4.17$ |
| Test error | $40.99 \pm 56.28$ | N/A | $2.74 \pm 9.94$ | $2.34 \pm 5.79$ |
| Prediction error | $37.87 \pm 117.02$ | N/A | $17.17 \pm 71.48$ | $23.95 \pm 66.61$ |
| **Task 2: Pendulum (embed)** | | | | |
| Model Parameter | 0.65M | 0.46M | 0.39M | 0.14M |
| Train error | $2.31 \pm 3.72$ | $0.59 \pm 1.634$ | $1.76 \pm 3.69$ | $0.067 \pm 0.276$ |
| Test error | $2.18 \pm 3.59$ | $0.49 \pm 1.762$ | $1.41 \pm 2.82$ | $0.052 \pm 0.241$ |
| Prediction error | $317.21 \pm 521.46$ | $14.31 \pm 29.54$ | $3.69 \pm 7.72$ | $0.20 \pm 0.49$ |
| **Task3: CartPole** | | | | |
| Model Parameter | 1.01M | 0.82M | 0.67M | 0.51M |
| Train error | $15.53 \pm 22.52$ | $0.45 \pm 0.37$ | $4.84 \pm 4.42$ | $1.78 \pm 1.81$ |
| Test error | $25.42 \pm 38.49$ | $1.20 \pm 2.67$ | $6.90 \pm 8.66$ | $1.89 \pm 1.81$ |
| Prediction error | $332.44 \pm 245.24$ | $52.26 \pm 73.25$ | $225.22 \pm 194.24$ | $11.41 \pm 16.06$ |
| **Task 4: Acrobot** | | | | |
| Model Parameter | 1.46M | 0.97M | 0.78M | 0.51M |
| Train error | $2.04 \pm 2.90$ | $2.07 \pm 3.72$ | $1.32 \pm 2.08$ | $0.25 \pm 0.39$ |
| Test error | $5.62 \pm 9.29$ | $5.12 \pm 7.25$ | $3.33 \pm 6.00$ | $0.28 \pm 0.48$ |
| Prediction error | $64.61 \pm 145.20$ | $26.68 \pm 34.90$ | $9.72 \pm 16.58$ | $2.07 \pm 5.26$ |

