# OpenReview forum: "Symplectic ODE-Net: Learning Hamiltonian Dynamics with Control"
_ICLR.cc/2020/Conference — Accept (Poster)_

### Official Review · AnonReviewer2 · 2019-10-22
**Official Blind Review #2**

**Rating:** 8

**Review:**

In this paper, the authors propose a framework for learning the dynamics of a system with underlying Hamiltonian structure subjected to external control. Based on the extended equations of motion, the authors suggest how to apply NeuralODE in a way that makes use of the prior information that the unconstrained system is Hamiltonian and subjected to a control term. For a range of tasks, the authors then demonstrate that the proposed SymODEN framework can learn the dynamics and recover the known analytical solution, and that they can derive a controller that allows them to drive the system to a target configuration.

I find this paper very interesting and the formulation elegant. In particular, I appreciate that the paper is pretty much self-contained and that the authors derive the theory from first principles. However, there are a couple of points (listed below) that should be addressed prior to publication to improve the clarity of the paper, and to help the reader to fully appreciate the depth of the experimental section. If these points were addressed in sufficient detail, I would be willing to increase the score:

1) Reading the abstract and the introduction, I got the impression that SymODEN can be applied to any physical system while the method is in fact only applicable to systems governed by Eq. (11). I think it’s worth mentioning in the text that not every physical system follows (constrained) Hamiltonian Dynamics.

2) You mention that a controller is designed based on the learned dynamics. For example, in the first sentence in Sec. 2.2: ‘Once the dynamics of a system have been learned, it can be used to synthesize a controller to maneuver the system to a reference configuration q*’. I think it is important to specify here which dynamics you are referring to, i.e. the constrained or unconstrained dynamics. Later on it becomes clear that you use constant-u training data and that u is part of the input to the model, so it has to be constrained dynamics, but at this stage it is still unclear to the reader.

3) In Eq. (11), you set du/dt = 0 without motivating this restriction. Can you provide at least one sentence on why this is an interesting choice and back it up with a reference? Furthermore, the sentence above Eq. (11) is broken and needs fixing.

4) Tasks (general): You consider a range of tasks and I appreciate that you start with a simple and intuitive system. However, I think a bit more guidance throughout the tasks section would be very helpful. In particular, I would suggest that you provide a short (one or two sentences) summary at the beginning of each task to say what exactly it is that you are trying to test or demonstrate. Furthermore, I would suggest showing the summary of  results, i.e. Sec. 4.2, after you introduce the individual tasks rather than before.

5) Task 2: This task addresses multiple things in one go: Initially, you demonstrate that you can recover the results of Task 1 without access to the generalised momenta, and explain why this can only be done up to a constant scaling factor. This is very interesting and clear. However, then you jump straight into the controller and things become a little unclear because, at this stage, it still seems that the dynamics are the same as in Task 1, i.e. unconstrained. Please add a sentence or two for clarification. Another important aspect that is not commented on at all is the behaviour of u(t) in Eq. (27). In particular, isn’t u(t) expected to satisfy du/dt = 0 based on Eq. (11)? The results in Fig. 6 suggest that this is not the case (see time interval [2, 6]). This seems like an interesting and surprising behaviour, especially because SymODEN was only trained with constant-u training data. I would appreciate if the authors could comment on this. I would also suggest to add horizontal lines to Fig. 6 to indicate the expected results.

6) Task 4: Why did you not explain this task in a dedicated section like you did for all other tasks?

7) Symplectic: Since both the method and the title of the paper contain the word ‘symplectic’, it would be good if you explained what the term actually means.

8) ‘Our results show that incorporation of such physics-based inductive bias can provide knowledge about relevant physical properties (mass, potential energy) and laws (conservation of energy)...’. To me, this statement is slightly misleading. You did not demonstrate that SymODEN ‘provides knowledge’ of laws of the system; energy conservation (for u = 0) as a law is hard-coded into your network. The specific value of the energy can be inferred but that I would consider a physical property. I would suggest to change the wording to reflect this clearly.

9) Introduce the acronym ODE much earlier than in Sec. 3.1.

10) Model training: What happens if you use unseen initial conditions rather than the ones in the training data? Perhaps you could add a comment to clarify.

11) There are many typos and grammar mistakes in the paper. Please revise it carefully. To give you a few examples:
‘are both reformulation’ -> ‘are both reformulations’
Sec 2.1: Decide on whether you use plural or singular for ‘dynamics’ and be consistent.
‘on a equal footing’ -> ‘on an equal footing’
‘beyond classical mechanics, the Hamiltonian’ -> ‘Beyond classical mechanics, Hamiltonian …’
‘Hamiltonian is same as’ -> ‘Hamiltonian is the same as’
‘represents potential energy’ -> ‘represents the potential energy’
‘trajectory actually converge to’ -> ‘trajectory actually converges to’
‘a ODE solver -> ‘an ODE solver’.
‘Lagrangian and Hamiltonian formulation’ -> ‘Lagrangian and Hamiltonian formulations’
‘assume that q and p evolves’ -> ‘assume that q and p evolve’
‘translational coordinate’ -> ‘translational coordinates’
‘naive baseline model approximate’ -> ‘naive baseline model approximates’
etc.


*************************************
The authors addressed my comments and answered  my questions clearly.  I therefore increased the score. *************************************

**Experience Assessment:**

I have published one or two papers in this area.

**Review Assessment: Checking Correctness Of Derivations And Theory:**

I assessed the sensibility of the derivations and theory.

**Review Assessment: Checking Correctness Of Experiments:**

I assessed the sensibility of the experiments.

**Review Assessment: Thoroughness In Paper Reading:**

I read the paper thoroughly.

---

> ### Author Response · Authors · 2019-11-13
> **Thanks so much for your constructive feedback!**
>
> Thank you for reviewing our paper! We greatly appreciate your insightful comments and constructive feedback.
>
> In the following, we address your concerns individually.
>
> 1) In fact, our method is applicable to the class of systems described by Eqn.4 instead of Eqn.11. We have used Eqn.11 only for training purposes. In our response to the next point, we have provided a detailed discussion to address this ambiguity. We agree that Eqn.4 cannot describe the dynamics of every physical system. However, Eqn.4 is inspired by the port-Hamiltonian systems which are applicable to a large class of systems since it takes dissipation and external forcing into account. In this current work, we only consider external forcing with a focus on the control of physical systems. Adding dissipation to accommodate a broader variety of systems will be the topic of future work.
>
> 2) We use the “constant $u$” dynamics (Eqn.11) only for training purposes. As the Neural ODE framework requires the dimension of the domain of the input function to be the same as the dimension of the corresponding co-domain/range, we need to use an augmented dynamics. Eqn.11 provides the simplest form of augmented dynamics, since it uses a “constant $u$”. Then, if we create a dataset of trajectories each of which correspond to different, but constant, values of $u$, we can use it to train the model by leveraging Eqn.11. Once we have a trained model, we can actually apply any time-varying input $u$ to the dynamics (Eqn.4).
> This is indeed an expected outcome. In the SymODEN framework, we are actually learning the functions $H(q, p)$ and $g(q)$. We can learn $H(q, p)$ by using a constant $u=0$. And $g(q)$ can be learned by using training data with a non-zero $u$. With different values of $u$, such as [-2.0, -1.0, 1.0, 2.0] we are able to learn the input matrix $g(q)$. If $u$ is multi-dimensional, we can create a training dataset by considering inputs in such a way that any given input in the dataset has only one non-zero component. For example, the trajectories which were created using inputs with non-zero entry at the $i$-th component, will help us learn the $i$-th column of the input matrix $g(q)$. Learning an accurate enough $H(q, p)$ and $g(q)$ ensures that we have also learned the dynamics with high accuracy. Afterwards, applying any “time-varying $u$” as an input to the system would not create any issue.
>
> 3) We’ve fixed the sentence above Eqn.11. As explained in our response to the previous point, this assumption on the training dataset aids the learning process. Also, in traditional system identification, constant external forcings are often used to get the system responses.
>
> 4) Short summary of each task has now been added to the beginning of each subsection. Hopefully it provides more guidance throughout the task section. We have also moved the summary of results (previously Section 4.2) to the end of Section 4.
>
> 5) We hope that our responses to point (2) and (3) have clarified the ambiguity. We use “constant $u$” only for training purposes. Once the dynamics have been learned, any “time-varying $u$” can be applied for prediction and control tasks. Also, in Figure 4 (previously Figure 6), we have included horizontal lines to highlight the expected results.
>
> 6) The Acrobot is also an underactuated system, which means that we can learn the dynamics with high accuracy but we cannot design a good controller by using only potential energy shaping. In particular, as $g(q)g(q)^T$ is not invertible in this case, we also need kinetic energy shaping in order to design a controller. Our future work will focus on how to incorporate kinetic energy shaping into a deep learning framework. However, we have also trained SymODEN on a fully actuated version of Acrobot. Due to restrictions on the manuscript length, we have included the corresponding results inside the Appendix (Appendix E: Fully-actuated Cartpole and Acrobot) and added a subsection in the main body to describe the Acrobot task (4.5 Task 4: Acrobot).
>
> 7) We have added a footnote to the abstract to give rationale for using the word “Symplectic”.
>
> 8) Thank you for directing our attention to this delineation. We have rephrased this sentence. Now it reads as follows: “Our results show that incorporation of such physics-based inductive bias offers insight about relevant physical properties of the system, such as inertia, potential energy, total conserved energy.”.
>
> 9) In the updated version of our paper, we have introduced the acronym ODE within the Abstract itself.
>
> 10) Figure 6 (in the updated version) shows MSE and Total energy of a trajectory with previously unseen initial conditions. In all our experiments, we have a separate test set to make sure our models generalize well. We have now included a new section in the Appendix (Appendix F: Test Errors of the Tasks) to show the test errors for all the tasks.
>
> 11) Thank you for pointing out to these typos, we have corrected them in the updated version.

---

### Official Review · AnonReviewer1 · 2019-10-24
**Official Blind Review #1**

**Rating:** 8

**Review:**

Update: I have read the author's response. Thank you!
***

This is an excellent paper that integrates inductive biases from physics into learning dynamical models that can then be integrated into deep RL-based control tasks.

The model approximates the dynamical function f(q, p, u), where q are the generalised coordinates of the system (mixture of positions in R^1 and angles in S^1), p are the generalised momenta and u is the external control input. Function f can then be integrated by a numerical solver, and used as the dynamical function in a Neural ODE (ordinary differential equation) for modelling the continuous time evolution of the nonlinear dynamical system. The dynamics function is explicitly written as the equations of the Hamiltonian dynamical system, involving the 1) inverse of the mass function, 2) potential energy and 3) control function, in a complex graph (Figure 1) that transforms positional and angular coordinates and momenta x and the external control into f(x, u).

The derivation of the method is long but very well written and didactic. The experiments on the control suite of OpenAI Gym  are simple (pendulum and cart-pole only) but thorough, and compare the proposed method with a non-inductive bias method (still relying on Neural ODEs), a simpler naive and geometric baselines. Overall, the paper is very easy to follow (even for someone who does not work on control experiments in deep RL) while addressing complex physics.

Minor remarks:
Can you define g in 2.1?
Can you add the derivation of equations (8) through (10) in the appendix?
The second to last sentence on page 4 seems unfinished.
Can the authors comment on how this model would scale to larger (e.g. multiple joints) dynamical systems?

**Experience Assessment:**

I have read many papers in this area.

**Review Assessment: Checking Correctness Of Derivations And Theory:**

I carefully checked the derivations and theory.

**Review Assessment: Checking Correctness Of Experiments:**

I assessed the sensibility of the experiments.

**Review Assessment: Thoroughness In Paper Reading:**

I read the paper thoroughly.

---

> ### Author Response · Authors · 2019-11-13
> **Thanks so much for your encouraging response!**
>
> Thank you for reviewing our paper! We greatly appreciate your insightful comments and constructive feedback. We have updated our paper to address your comments/concerns.
>
> In the following, we address your concerns individually:
>
> --- We have updated our paper to include a definition of $g(q)$ after Equation 4 in Section 2.1.
>
> --- We have now included a new section in the Appendix (Appendix B: Special Case of Energy-based Controller - PD Controller with Energy Compensation) to show derivations of these equations describing external control .
>
> --- We have fixed this typo.
>
> --- To incorporate physics-based prior knowledge while learning dynamics from observed time-series data, our framework (SymODEN), in essence, learns three functions $-$ $M^{-1}(q)$, $V(q)$ and $g(q)$. We have shown that with a small dataset, containing as low as 16 training trajectories with 20 time steps in each, we can learn accurate models for simple systems and design controllers based on the learned model. For larger dynamical systems, we believe that as long as the underlying dynamics is governed by Equation 4 and large enough neural networks are used to learn those three functions, our framework should work quite well. For example, Equation 4 can describe the dynamics of any large mechanical system whose kinematic structure can be represented by kinematic trees. However, as $M^{-1}(q)$, $V(q)$ and $g(q)$ typically involve more parameters for larger dynamical systems, this might require more data to train the model. Still, compared with the baseline models and model-free reinforcement learning approaches, the amount of data required in our framework should be smaller.

---

> ### Author Response · Authors · 2019-11-14
> **Uploaded a Revision with Changes Highlighted**
>
> We have now uploaded a revised version of our manuscript with all the changes highlighted. Moreover, this revision includes additional pointers (shown within red boxes in the right margin)  to connect key changes to specific comments from your review. We are using the format “R1-C$x$” to show the connection between your Comment-$x$ and the corresponding change in the manuscript.
>
> *** Please note that these pointers are only visible in offline PDF-viewers.

---

### Official Review · AnonReviewer3 · 2019-10-25
**Official Blind Review #3**

**Rating:** 6

**Review:**

============ Update ===========
The authors have done a good job at addressing my concerns, and the revised version of the submission is substantially improved. I have adjusted my score and recommend accepting the paper.
==============================

Recent work has explored encoding analytic mechanics formulae into neural networks as inductive biases to learn physics models that generalize better. Neural networks are implemented to learn quantities like kinetic and potential energy rather coordinate derivatives. In this paper, the work of [1] is extended to incorporate a more generalizable approach to modeling functions on angles, integral approach where errors are backpropagated through an ODE solver rather than fitting errors in the derivatives, and modeling response to controls.

For Lagrangian and Hamiltonian systems it’s often easier to work with non-euclidean generalized coordinates rather than a constrained Euclidean system, however it can be difficult to design well parametrized neural network functions on a manifold like a circle. The authors address this by still expressing the having the Hamiltonian expressed in terms of circular generalized coordinates, but parametrizing the functions on the Euclidean embeddings. The paper shows that this approach does not have a problem with generalizing to large angles that the naïve approach does.

The integral approach to computing errors seems sensible, and appears to work well but no comparison is made to the previous method working with derivatives. This would be useful in demonstrating that the predictive performance is at least no worse than the approach taken in [1] and doesn’t require knowing or estimating derivatives. Also it would be good to have an ablation study investigating predictive performance as a function of tau, the number of integration timesteps for computing the error.

The paper shows evaluation of the learned dynamics system for control on two examples, an inverted pendulum and CartPole. In the inverted pendulum example, the learned potential energy and the control response is used to design a control that shapes the potential energy and with additional damping. Since the control output is closely related to a PD controller, it would be good to compare against a standard P(I)D controller that doesn’t depend on the model. For the CartPole system, the control is exactly a PD controller and it’s not clear how the Hamiltonian/dynamics model are used at all in this example. Generally the paper does a good job at demonstrating benefits in modeling ability and generalization, but the experiments applying this model to control are not very convincing. Having an example where the model is applied for a standard approach like MPC for one of these problems would useful for gauging efficacy in possible control applications.

There are some promising leads explored in this paper for learning physical system dynamics effectively with neural nets. I think there is a lot of promise in the approach, however some of the improvements over past work have not been adequately tested (integral approach and sensitivity to # of integration steps) and the control experiments are not very convincing although it seems like they could be. I lean towards a weak reject for the paper as is, but if the authors flesh out the control experiments and do some ablation studies I will improve my score.

Minor Comments and Questions:
Is f tilde in equation 11 parametrized to have 0 output on u dot or is it expected that this relationship would simply be approximated by the model?

[1] Sam Greydanus, Misko Dzamba, and Jason Yosinski. Hamiltonian Neural Networks. arXiv:1906.01563, 2019.


**Experience Assessment:**

I have read many papers in this area.

**Review Assessment: Checking Correctness Of Derivations And Theory:**

I carefully checked the derivations and theory.

**Review Assessment: Checking Correctness Of Experiments:**

I carefully checked the experiments.

**Review Assessment: Thoroughness In Paper Reading:**

I read the paper thoroughly.

---

> ### Author Response · Authors · 2019-11-13
> **Thanks so much for your constructive feedback!**
>
> Thank you for reviewing our paper! We greatly appreciate your insightful comments and the super constructive feedback. We have updated our paper to address your comments/concerns.
>
> To obey length restrictions, we have split our official response into two parts. This is Part 1.
>
> In the following, we address your concerns individually.
>
> Ablation Study of Differentiable ODE Solver:
> We have carried out the ablation study to distinguish the effect of using a differentiable ODE solver. We compare the result of Unstructured SymODEN and HNN since both models approximate the Hamiltonian using a neural network and the main difference is the use of differentiable ODE Solver. As HNN does not employ any angle-aware design, we perform the ablation study only for Task 1. We have discussed the details of the experimental setup and the results in a new section in the Appendix (Appendix C: Ablation Study of Differentiable ODE Solver). We can see that Unstructured SymODEN performs significantly better than HNN in terms of training and prediction errors associated with our experiment (Task 1: Pendulum with Generalized Coordinate and Momentum Data). From the evolution of MSE for a particular trajectory (Figure 7), we can notice that Unstructured SymODEN makes better prediction. In fact, the HNN loss term compares the estimated and true symplectic gradient while SymODEN loss term compares the estimated trajectories. Therefore, even if the training error from the HNN is comparable to that from the Unstructured SymODEN, the error in HNN is accumulated during trajectory prediction (which is obtained by integrating the symplectic gradient). This, as expected, leads to a larger error in predictions from HNN.
>
> Effects of $\tau$:
> We have added a new section in the Appendix (Appendix D: Effects of the time horizon $\tau$) to discuss the effects of time horizon $\tau$. From these results, we can see that larger values of $\tau$ lead to smaller error. This is expected since a large $\tau$ penalizes inaccurate predictions in the long run. We have also observed in our experiments that a larger $\tau$ requires more time to train the model since it involves more function evaluations.
>
> PD controller:
> In Section 2.2 and “Appendix B: Special Case of Energy-based Controller - PD Controller with Energy Compensation” (in the updated version), we have shown that if the desired potential energy is given by a quadratic form, the “potential energy shaping + damping injection” becomes equivalent to a “PD controller + energy compensation”. A pure PD controller would work well only around the equilibrium point since in that region the linearized dynamics matches the true dynamics.
>
> Control of CartPole system:
> Thanks for highlighting that the PD controller design for the CartPole system does not depend on the dynamics. The CartPole system is an underactuated system; incorporating deep learning into controller design for underactuated systems would be the focus of our future work. However, we have also trained SymODEN on a fully-actuated version of CartPole. Due to restrictions on the manuscript length, we have included the corresponding results inside Appendix (Appendix E: Fully-actuated Cartpole and Acrobot). Here we have shown that the SymODEN framework can learn the dynamics and control the fully-actuated CartPole. We have also removed the PD controller results for CartPole system since the learned model has not been used to design this controller (as you have pointed out).
>
> MPC:
> We considered performing MPC based on the learned dynamics during the course of our research on this topic. In particular we used mpc.pytorch [1]. However, it didn’t perform as expected in these control tasks. The main reason might be the fact that the dynamics learned by a neural network is indeed an approximation (albeit a very accurate one); as mentioned in the mpc.pytorch documentation [2] - “Sometimes the controller does not run for long enough to reach a fixed point, or a fixed point doesn’t exist, which often happens when using neural networks to approximate the dynamics. When this happens, our solver cannot be used to differentiate through the controller, because it assumes a fixed point happens.” Therefore, we switch towards energy-based controllers since we are learning the energy and it’s natural to leverage the learned energy for controller design. In Appendix E (Fully-actuated Cartpole and Acrobot), we trained our model on the fully-actuated version of Cartpole and Acrobot and have shown that our framework can successfully control those fully-actuated systems. We hope our work creates new research directions combining MPC or energy shaping control with interpretable end-to-end learning framework.
>
> [1] Amos, Brandon, et al. "Differentiable MPC for End-to-end Planning and Control." Advances in Neural Information Processing Systems. 2018.
> [2] https://locuslab.github.io/mpc.pytorch/
>
> *************** Part 1 of the Official Response ***************

---

> > ### Author Response · Authors · 2019-11-14
> > **Uploaded a Revision with Changes Highlighted**
> >
> > We have now uploaded a revised version of our manuscript with all the changes highlighted. Moreover, this revision includes additional pointers (shown within red boxes in the right margin) to connect key changes to specific comments from your review. We are using the word “R3” to show the connection between your Comments and the corresponding changes in the manuscript.
> >
> > *** Please note that these pointers are only visible in offline PDF-viewers.

---

> ### Author Response · Authors · 2019-11-13
> **Part 2 of the Official Response**
>
> *************** Part 2 of the Official Response ***************
>
> Equation 11: $\tilde{f}$ is parametrized to have 0 output on $\dot{u}$. Eqn.11 is only used for training when we have trajectory data with “constant $u$”. We can safely parametrize $\dot{u}=0$ since this is consistent with the data. Once the model has been trained, we can apply a time-varying $u$ as we have done in the control tasks.
>
> ********** See Below for Part 1 of the Official Response **********

---

### Public Comment · ~Yiping_Lu1 · 2019-09-27
**Related works**

Congrats on your work and I really enjoy reading it.
I'm writting the comment to introduce some of our related works on discretization Neural ODEs and Control
Lu Y, Zhong A, Li Q, et al. Beyond finite layer neural networks: Bridging deep architectures and numerical differential equations[J]. arXiv preprint arXiv:1710.10121, 2017.
Long Z, Lu Y, Ma X, et al. PDE-net: Learning PDEs from data[J]. arXiv preprint arXiv:1710.09668, 2017.

Also these early papers also aim to connect ODEs and deep learning
Weinan, E. "A proposal on machine learning via dynamical systems." Communications in Mathematics and Statistics 5.1 (2017): 1-11.
Li, Qianxiao, et al. "Maximum principle based algorithms for deep learning." The Journal of Machine Learning Research 18.1 (2017): 5998-6026.
Weinan, E., Jiequn Han, and Qianxiao Li. "A mean-field optimal control formulation of deep learning." Research in the Mathematical Sciences 6.1 (2019): 10.
Haber, Eldad, and Lars Ruthotto. "Stable architectures for deep neural networks." Inverse Problems 34.1 (2017): 014004.
Chang, Bo, et al. "Multi-level residual networks from dynamical systems view." arXiv preprint arXiv:1710.10348 (2017).
Ruthotto, Lars, and Eldad Haber. "Deep neural networks motivated by partial differential equations." arXiv preprint arXiv:1804.04272 (2018).

---

### Decision · Program_Chairs · 2019-12-19

**Decision:**

Accept (Poster)

**Comment:**

This paper proposes a novel method for learning Hamiltonian dynamics from data. The data is obtained from systems subjected to an external control signal. The authors show the utility of their method for subsequent improved control in a reinforcement learning setting. The paper is well written, the method is derived from first principles, and the experimental validation is solid. The authors were also able to take into account the reviewers’ feedback and further improve their paper during the discussion period. Overall all of the reviewers agree that this is a great contribution to the field and hence I am happy to recommend acceptance.